# Chiral *C*_2_-Symmetric Diimines with 4,5-Diazafluorene Units

**DOI:** 10.3390/molecules24173186

**Published:** 2019-09-02

**Authors:** Eugene S. Vasilyev, Sergey N. Bizyaev, Vladislav Yu. Komarov, Yury V. Gatilov, Alexey V. Tkachev

**Affiliations:** 1N. N. Vorozhtsov Novosibirsk Institute of Organic Chemistry, Siberian Branch of Russian Academy of Sciences, 9 Academician Lavrentiev Ave., Novosibirsk 630090, Russia; 2Nikolaev Institute of Inorganic Chemistry, Siberian Branch of Russian Academy of Sciences, Novosibirsk 630090, Russia; 3Department of Natural Sciences, Novosibirsk State University, Novosibirsk 630090, Russia

**Keywords:** pinocarvone oxime, 4,5-diazafluorenon-9-one, chiral dipinodiazafluorenone, aromatic diamines, stable chiral diimines

## Abstract

A synthetic approach to a new group of stable chiral *C*_2_-symmetric diimines with the 4,5-diazafluorene core has been developed based on condensation of dipinodiazafluorene with aromatic diamines. The chemical structures of new compounds were proven by spectroscopic methods and X-ray crystallography. All the compounds form solvates with organic solvents (chloroform, benzene, 1,4-dioxane) and water. Specific spectral data of the new compounds are explained using calculated data (DFT). Diimines of the pinodiazafluorene series give colored reactions with transition metal ions and can be regarded as prospective polydentate ligands with interesting luminescent and chiroptical properties.

## 1. Introduction

Various 4,5-diazafluorene derivatives are of interest as components of photoelectrochemical cells [1,2], fluorescent sensors [3], as ligands for catalytic reactions [4,5], and are considered potential bioactive compounds [6]. In previous works 4,5-diazafluoren-9-one, which is prepared from 1,10-phenanthroline, was employed as starting material. However, syntheses of substituted 4,5-diazafluorene derivatives involves many steps [7]. Recently we managed to obtain new chiral dipinodiazafluorenone **1** ((1*R*,3*R*,8*R*,10*R*)-2,2,9,9-tetramethyl-3,4,7,8,9,10-hexahydro-1*H*-1,3:8,10-dimethanocyclopenta [1,2-*b*:5,4-*b*′]diquinolin-12(2*H*)-one) in two steps from (+)-pinocarvone oxime [8]. Molecule **1** has a 4,5-diazafluoren-9-one core fused to two nopane moieties. In this paper, we investigated the reactions of the chiral dipinodiazafluorenone **1** with diamines. Diamines are known to react with unsubstituted 4,5-diazafluoren-9-one to produce diimines with low solubility in common organic solvents [9]. We thus investigated the condensations of chiral dipino-diazafluorenone **1** with aromatic diamines (4,4′-diaminodiphenylmethane, 3,3′,5,5′-tetramethyl-benzidine, *ortho*-, *meta*- and *para*- phenylenediamines, 1,5-diaminonaphthalene, and 1,8-diamino-naphthalene) and hydrazine and found that the reaction produces quite stable diimines in moderate and good yields (Scheme 1).

## 2. Results and Discussion

Boiling a mixture of amine and ketone in acetic acid is a common method for the preparation of diimines [1]. However, in the case of dipinodiazafluorenone **1**, this method provides perceptible yields only in the condensation with 4,4′-diaminodiphenylmethane or 3,3′,5,5′-tetramethylbenzidine, and the corresponding diimines **2** and **3** can be isolated in 27% and 13% yields, respectively. Condensation with other diamines provided the desired products in low yields (if at all), wherein the corresponding monoimines were the main products. Transformation of ketones to diimines in the reaction with diamines is obviously a two-step process, and in the case of fully conjugated diamines, the nucleophilicity of the intermediate monoimine is insufficient for condensation with the second ketone molecule. Besides boiling with acetic acid, we have checked various other condensation systems (TsOH–benzene, TsOH–toluene, TsOH–xylene, BF_3_ × Et_2_O–toluene) and found TiCl_4_–Py to be the best choice (Table 1).

Condensation of dipinodiazafluorenone **1** with 1,8-diaminonaphthalene under various conditions (AcOH, TsOH, BF_3_ × Et_2_O, TiCl_4_–Py) usually produced a complex mixture of colored air-sensitive products, spirocyclic compound **9** with a 2,2-disubstituted 2,3-dihydro-1*H*-perimidine moiety being the only isolatable component. The air-sensitive compound **9** may be of independent interest, so this compound was prepared in moderate yield by the condensation with HClO_4_–EtOH in an argon atmosphere. Formation of spirocompound **9** can be explained in terms of preferable intramolecular acid-catalyzed cyclization of the intermediate monoimine rather than its reaction with the second dipinodiazafluorenone molecule (Scheme 2). It is noteworthy that when condensed with *ortho*-phenylenediamine, the spiro compound **10** of a similar architecture was not detected.

All the synthesized compounds are crystalline substances and crystallize as solvates with solvents (organic solvents and water), whose signals are always detected in NMR spectra. The solvates are unstable at room temperature and are rapidly destroyed in air. For this reason, it was not possible to obtain reproducible microanalysis and melting point results. However, in some cases, it was possible to prepare crystals, for which low temperature X-ray diffraction analyses were carried out (see below) to prove chemical structure of the new diimines. All the compounds produce molecular ions M^+•^ upon electron impact ionization, and the exact masses of the molecular ions according to high resolution mass spectrometry are in good agreement with the calculated ones. Purity of the compounds is confirmed by ^1^H-NMR spectroscopy: only signals of the target substances and solvate solvents are detected in the spectra, whereas signals of impurities are at the level of the C-13 satellites of the main substance signals, which indicates that the purity of all new compounds is about 99%.

All NMR spectra of compounds **2**–**8** have two sets of signals belonging to unequivalent pinopyridine moieties, the chemical shifts of the H-10′ and H-1′ signals being the most sensitive to the character of the substituent at the imine nitrogen because of additional shielding due to strong magnetic anisotropy of aromatic rings. In molecule **3**, restricted rotation around the N–Ar single bond due to the presence of two methyls at the *ortho*- and *ortho*′-postions of the benzene ring leads to new prochiral axis in the molecule resulting in the formation of pairs of diastereotopic groups with unequivalent chemical shifts (**15** and **15**′, **16** and **16**′, **18** and **18**′). Transformation of dipinodiaza-fluorenone **1** into a Schiff base results in loss of *C*_2_-symmetry of the fused diazafluorene moiety leading to nonequivalence of the two pinopyridine frames (Scheme 3).

Vibrational spectra of all the synthesized compounds have the same most intense line at 1392–1398 cm^−1^, like in the spectra of the simplest derivatives of this series [8]. According to the quantum chemical calculations (DFT BPE0/def2-TZVP), this line appears due to a series of the specific skeletal vibrational modes of the dipinodiazafluorene moiety. There are several oscillations with the same type of valence vibrations in the diazafluorene core and different character of deformations in the pinane fragment (Figure 1). These oscillations have similar frequencies (Δν = 1–10 cm^−1^) and in the real spectrum merge into one wide intensive asymmetric-shaped band.

The displacements shown do not involve the C-14 atom, so substituents at C-14 carbon have little effect on the frequency and intensity of the corresponding IR band, and for this reason, an absorption band of the same intensity appears at 1392–1398 cm^−1^ in the IR spectra of all compounds of this series.

The mass of crystals of compound **7** was analyzed for the selection of crystals suitable for X-ray study. The bulk mass of crystals is inappropriate for single crystal X-ray diffraction, however, it was possible to select several well-formed crystals, but the results of the analysis showed that these crystals consist of molecules of structure **11** (Figure 2). 

HRMS and ^1^H-NMR of the selected crystals are in agreement with structure **11**. It is not possible to estimate the yield of this compound, because in the ^1^H-NMR spectrum of compound **7** the signals of product **11** are not visible against the background of the signals of impurities whose content is less than 0.5%. Compound **11** could arise from chlorination of the starting compound (1,5-diaminonaphthalene) followed by condensation with dipinodiazafluorenone **1**. Formation of compound **11** is also possible as a result of chlorination of the corresponding intermediate monoimine.

Chemical structures and stereochemistry of the molecules **4**, **6**, **8** and **11** were proven by X-ray single crystal diffractometry (Figure 3, Figure 4, Figure 5 and Figure 6). Molecules **6** and **8** are located on the C_2_ axis, passing through the middle of the N-N bonds in **8** and C(14)-C(14), C(16)-C(16) in **6**. Molecules **4** and **11** are located in general positions. The geometry of dimetanocyclopentadiquinoline fragments is close to similar data [8]. Molecules **4** and **8** are almost flat, whereas molecule **6** has a tweezer shape. The angles between the planes of cyclopentadipyridines in the structures **4**, **6** and **8** are 6.9, 58.0 and 15.3°, respectively. The linking benzene ring in **4** is tilted off average plane of the molecule by 76.0°. A similar tilt of the “naphtalene” plane with an angle of 65.9° is observed in the partial substitution product **11**. In the structure of **11**, formation of intermolecular hydrogen bonds are probable between N atoms of diazaflourene fragment and hydrogens of the primary amino group (N(1a)… N(13) 3.00 Å).

All the compounds studied by X-ray single crystal diffractometry are solvates. In the crystals of **4** and **6**, the N(1*x*) atoms are bonded to the chloroform by hydrogen bonds C-H…N (e.g., for **6** H…N 2.48, 2.49 Å, C-H…N 148, 161°). Molecules of compound **8** in the crystal are packed so that free solvent accessible volume derived from PLATON [10,11] routine analysis was found to be 43.8% (1321.9 Å^3^). This volume is occupied by highly disordered solvent molecules. We employed PLATON/SQUEEZE procedure to calculate the contribution to the diffraction from the solvent region. Compound **11** forms a solvate with acetonitrile.

Having the same absolute configuration of nopinane annelated moiety, bis-imines **2**–**8** have very different values of optical activity measured at 589 nm (${\left[\alpha \right]}_{D}$ from +5.9 to −249) and a great difference in optical rotation at 589 and 546 nm, which indicates the presence visible-light optically active absorption bands for all the synthesized bis-imines. When exposed to ultraviolet radiation, chloroform solutions of some bis-imines exhibited photoluminescence: violet blue for **2** and **4** and greenish blue for **6** with maxima at 402, 394 and 397 nm respectively (Figure 7). Bisimines **2–8** give color reactions when their TLC spots are sprayed with methanol solutions of transition metal salts at room temperature (Table 2). The instant development of color indicates the rapid formation of chelate complexes, while the different color in the treatment of compounds **2–8** with the same metal salt suggests a variety of structures of the complexes formed.

## 3. Materials and Methods

### 3.1. General Information

All the organic solvents used were freshly distilled. Merck silica gel 60 (0.063–0.100 mm) was used for preparative column chromatography. Analytical TLC was carried out on a ready-to-use plates (SiO_2_ on Al foil, visualization by spraying with a solution of FeCl_3_ × 6H_2_O, (NH_4_)_2_Ce(NO_3_)_6,_ CuCl_2_ × 2H_2_O, or Cu(ClO_4_)_2_ × 6H_2_O–see Table 2).

NMR spectra were recorded at 25–28 °C for solutions (*c* 20–40 mg/mL) on a DRX-500, Avance 400 or Avance 300 spectrometer (Bruker Corporation, Billerica, MA, USA) locked to the deuterium resonance of the solvent. The chemical shifts were calculated relative to the solvent signals (CDCl_3_) used as the internal standard: δ*_C_* 76.90 ppm and δ*_H_* 7.24 ppm.

IR spectra were recorded on a Bruker TENSOR 27 spectrophotometer in KBr (*c* 0.25%). UV-spectra were recorded on an 8453 instrument (Agilent Technologies, Inc., Santa Clara, CA, USA). Optical rotation was measured on a PolAAr 3005 polarimeter (Optical Activity Ltd., Ramsey, Cambridgeshire, United Kingdom). Mass spectra were obtained on a Thermo Electron DFS (electron impact ionization, EI, 70 eV) mass spectrometer (Thermo Electron GmbH, Bremen, Germany). Excitation and luminescence spectra for chloroform solutions were detected using a Cary Eclipse spectrofluorimeter (Varian Australia Pty Ltd., Mulgrave, Victoria, Australia) at room temperature with averaging time of 0.1–0.5 s.

X-ray crystallographic data were obtained on a Bruker Kappa Apex II (compounds **6** and **8**) and Bruker Apex Duo (compounds **4** and **11**) diffractometers with CCDs using graphite-monochromated MoKα radiation (λ = 0.71073 Å). Experimental data reduction was performed using APEX2 suite [12]. The structures were solved by direct methods and refined by the full-matrix least-squares technique against *F*^2^ in the anisotropic-isotropic approximation. The H atoms positions were calculated with the riding model. All calculations were performed using SHELXTL-2014/7 [13] assisted with Olex2 GUI [14]. 

Geometry optimization and vibrational frequencies at the DFT level were obtained using ORCA program system [15] with the hybrid exchange-correlation functional PBE0 and the valence triple-zeta polarization basis set def2-TZVP [16].

### 3.2. Starting Materials

(1*R*,3*R*,8*R*,10*R*)-2,2,9,9-Tetramethyl-3,4,7,8,9,10-hexahydro-1*H*-1,3:8,10-dimethanocyclopenta [1,2-*b*:5,4-*b*′]diquinolin-12(2*H*)-one (**1**) as yellow crystals with mp 247 °C (benzene–EtOAc 1:1) and ${\left[\alpha \right]}_{D}^{30}$ − 224 (*c* 0.59, CHCl_3_) was prepared from (–)-α-pinene (Fluka Chemie GmbH, Buchs, Switzerland, Product Number 8060, 93% *e.e.*) via pinocarvone oxime as described earlier [8]. All the other chemicals used were commercially available.

### 3.3. Synthetic Procedures and Spectral Data

#### 3.3.1. Preparation of bis-imines with TiCl_4_-Py

A solution of TiCl_4_ (1.8 mmol, 200 µL) in THF (3 mL) was added dropwise at room temperature (+25 °C) to a solution of pinodiazofluorenone **1** (1 mmol, 370 mg) and the corresponding aromatic diamine (0.5 mmol) in THF (2 mL) followed by addition of pyridine (2 mL). The reaction mixture was stirred at room temperature for 2 days. Afterwards mixture was percolated on silica gel (eluent–THF) and crystallized from acetonitrile–1,4-dioxane (1:1 *v*/*v*) to provide compounds **2** (361 mg, 80%), **3** (387 mg, 82%), **4** (337 mg, 83%), **5** (325 mg, 80%), **6** (317 mg, 78%), or **7** (302 mg, 74%).

#### 3.3.2. Condensation in Acetic Acid

A solution of pinodiazofluorenone **1** (370 mg, 1 mmol) and the corresponding aromatic diamine (0.5 mmol) in glacial acetic acid (5 mL) was heated at +110 °C for 4 h. The solvent was distilled off in vacuum. The residue was dissolved in dry acetonitrile (30 mL) followed by addition of Na_2_CO_3_ (4 g). The mixture was stirred for 20 min and filtered and the filtrate was concentrated under reduced pressure to leave the crude product (brown viscous oil), which was then purified by column chromatography (SiO_2_, CHCl_3_) to afford bis-imine **2** (27%), **3** (13%) or **4** (6%).

#### 3.3.3. Condensation with BF_3_ × Et_2_O

BF_3_ × Et_2_O (100 µL, 0.80 mmol) was added dropwise to a solution of pinodiazofluorenone **1** (244 mg, 0.66 mmol) and the corresponding aromatic diamine (0.33 mmol) in toluene (3 mL). The reaction mixture was heated at +110 °C for 4 h, cooled down to room temperature and diluted with THF (30 mL). Powdered Na_2_CO_3_ (5 g) and NaF (5 g) was then added, the resultant mixture was stirred for 1 h and filtered. The filtrate was concentrated under reduced pressure; the residue was chromatographed on a silica gel column (CHCl_3_) to give bis-imine **4** (72 mg, 0.089 mmol) in 27% yield.

#### 3.3.4. Condensation with benzene-TsOH

A mixture of pinodiazofluorenone **1** (1 mmol, 370 mg), hydrazine hydrate (0.5 mmol, 25 mg), *p*-toluenesulfonic acid (0.1 mmol, 17.2 mg) and benzene (60 mL) was stirred at reflux for 5 h. The reaction mixture was cooled, washed with aqueous ammonia, solvent was evaporated. Crude product was purified by column chromatography (eluent: benzene-chloroform). The reaction mixture was cooled down to room temperature, washed with 10% aqueous ammonia (2 × 10 mL), dried over Na_2_SO_4_ and concentrated under reduced pressure. The crude product was purified by column chromatography on a silica gel column (benzene-chloroform) to afford bis-imine **8** (203 mg, 50%).

#### 3.3.5. Condensation with toluene-TsOH

A mixture of pinodiazofluorenone **1** (0.81 mmol, 300 mg), corresponding aromatic diamine (0.40 mmol), *p*-toluenesulfonic acid (0.058 mmol, 10.0 mg) and toluene (10 mL) was stirred at reflux for 4 h. The solvent was distilled off in vacuum, the residue was taken up in acetonitrile and the resultant solution was stirred with powdered Na_2_CO_3_ (4 g) for 20 min at room temperature. The mixture was filtered and the filtrate was evaporated under reduced pressure to leave the crude product as brown viscous oil which was then chromatographed on a silica gel column (gradient elution, 0→100% *v*/*v* chloroform in benzene) to afford bis-imines **5** (154 mg, 47%) or **6** (66 mg, 20%).

#### 3.3.6. Condensation with xylene-TsOH

The reaction of pinodiazofluorenone **1** (300 mg, 0.81 mmol) with *o*-phenylenediamine (43 mg, 0.40 mmol) was carried out as described above for the transformation of pinodiazofluorenone **1** with toluene-TsOH, but *o*-xylene was used instead of toluene. Column chromatography of the crude product (SiO_2_, gradient elution, 0→100% *v*/*v* chloroform in benzene) yielded bis-imine **6** (164 mg, 50%).

### 3.4. Spectral Data

*4,4′-Methylenebis(N-((1R,3R,8R,10R)-2,2,9,9-tetramethyl-3,4,7,8,9,10-hexahydro-1H-1,3:8,10-dimethano-cyclopenta[1,2-b:5,4-b′]diquinolin-12(2H)-ylidene)aniline)* (**2**). Solvate with CHCl_3_ and H_2_O. Orange powder; ${\left[\alpha \right]}_{D}^{23}$ − 79 (*c* 0.52, CHCl_3_); ${\left[\alpha \right]}_{546}^{23}$ − 116 (*c* 0.52, CHCl_3_); UV (CHCl_3_)λ_max_ (lg ε) 393 (3.63), 339 (4.57), 332 (4.49), 324 (4.49), 250 (4.87) nm; IR (KBr) ν_max_ 3040–2850 (ν_C-H_), 1644 (ν_C=N_), 1598, 1565, 1555, 1500, 1468, 1424, 1395 (the most intensive band), 1329, 1262, 1217, 1181, 1102, 1072, 948, 927, 900, 834, 752 (δ_Ar–H_) cm^−1^; ^1^H-NMR (CDCl_3_, 400 MHz) *δ* 0.53 (3H; s, H8′), 0.69 (3H; s, H8), 0.98 (1H, d, *J* = 9.7 Hz, *pro*-*R*-H7′), 1.26 (3H, s, H9′), 1.29 (1H, d, *J* = 9.7 Hz, *pro*-*R*-H7), 1.41 (3H, s, H9), 1.78 (br. s, H_2_O), 2.23 (1H, dddd, *J* = 5.7, 5.4, 2.4, 2.4 Hz, H5′), 2.29 (1H, dd, *J* = 5.6, 5.6 Hz, H1′), 2.37 (1H, dddd, *J* = 5.7, 5.4, 2.4, 2.4 Hz, H5), 2.43 (1H, ddd, *J* = 9.7, 6.0, 5.7 Hz, *pro*-*S*-H7′), 2.71 (1H, ddd, *J* = 9.7, 6.0, 5.7 Hz, *pro*-*S*-H7), 2.85 (1H, dd, *J* = 5.6, 5.6 Hz, H1), 3.13 (1H, dd, *J* = 18.7, 2.8 Hz, H4′a), 3.17 (1H, dd, *J* = 18.7, 2.4 Hz, H4′b), 3.28 (2H, d, *J* = 2.6 Hz, H4), 4.09 (1H, s, H18), 6.23 (1H, s, H10′), 6.91 (2H, d, *J* = 8.2 Hz, H15), 7.24 (s, CHCl_3_), 7.26 (2H, d, *J* = 8.2 Hz, H16), 7.68 (1H, s, H10); ^13^C-NMR (CDCl_3_, 100 MHz) *δ* 21.1 (C-8 or C-8′), 21.2 (C-8′ or C-8), 25.8 (C-9 or C-9′), 25.9 (C-9′ or C-9), 31.7 (C-4 or C-4′), 31.8 (C-4′ or C-4), 36.9 (C-7 or C-7′), 37.1 (C-7′ or C-7), 39.2 (C-6 or C-6′), 39.4 (C-6′ or C-6), 39.6 (C-1 or C-1′), 39.8 (C-1′ or C-1), 40.9 (C-18), 46.8 (C-5 or C-5′), 46.9 (C-5′ or C-5), 77.1 (CHCl_3_), 118.6 (C-15), 123.4 (C-11′), 127.3 (C-10′), 129.5 (C-11), 129.6 (C-16), 130.4 (C-10), 137.9 (C-17), 141.0 (C-2 or C-2′), 142.4 (C-2′ or C-2), 149.5 (C-14), 158.1 (C-13), 159.7 (C-12 or C-12′), 159.9 (C-12′ or C-12), 161.0 (C-3 or C-3′), 161.5 (C-3′ or C-3); EI MS *m*/*z* 902 [M]^+^ (100), 859 (10), 551 (24), 550 (58), 451 (10), 370 (17), 355 (13), 207 (18); HREIMS *m*/*z* 902.5034 (calcd for C_63_H_62_N_6_^+•^, 902.5030).

*3,3′,5,5′-Tetramethyl-N4,N4′-bis((1R,3R,8R,10R)-2,2,9,9-tetramethyl-3,4,7,8,9,10-hexahydro-1H-1,3:8,10-dimethanocyclopenta[1,2-b:5,4-b′]diquinolin-12(2H)-ylidene)-[1,1′-biphenyl]-4,4′-diamine* (**3**). Solvate with CHCl_3_ and H_2_O. Yellow powder; ${\left[\alpha \right]}_{D}^{23}$ + 14 (*c* 0.29, CHCl_3_); ${\left[\alpha \right]}_{546}^{23}$ + 40 (*c* 0.29, CHCl_3_);UV (CHCl_3_) λ_max_ (lg ε)440 (3.32), 339 (4.43), 332 (4.36), 323 (4.42), 291 (4.58), 249 (4.69) nm; IR (KBr) ν_max_ 3040–2850 (ν_C-H_), 1647 (ν_C=N_), 1598, 1570, 1556, 1463, 1424, 1395 (the most intensive band), 1261, 1215, 1072, 947, 928, 862, 809, 774, 677 cm^−1^; ^1^H-NMR (CDCl_3_,500 MHz) *δ* 0.58 (3H, s, H8 or H8′), 0.75 (3H, s, H8′ or H8), 1.17 (1H, d, *J* = 9.7 Hz, *pro*-*R*-H7′), 1.31 (3H, s, H9 or H9′), 1.33 (1H, d, *J* = 9.7 Hz, *pro*-*R*-H7), 1.44 (3H, s, H9′ or H9), 1.62 (br.s, H_2_O), 2.09 (3H, s, H18 or H18′), 2.12 (3H, s, H18′ or H18), 2.29 (1H, dddd, *J* = 5.7, 5.4, 2.4, 2.4 Hz, H5′), 2.40 (1H, dddd, *J* = 5.7, 5.4, 2.4, 2.4 Hz, H5), 2.43 (1H, dd, *J* = 5.6, 5.6 Hz, H1′), 2.58 (1H, ddd, *J* = 9.7, 6.0, 5.7 Hz, *pro*-*S*-H7′), 2.74 (1H, ddd, *J* = 9.7, 6.0, 5.7 Hz, *pro*-*S*-H7), 2.90 (1H, dd, *J* = 5.6, 5.6 Hz, H1), 3.21 (2H, dd, *J* = 2.6, 1.5 Hz, H4′), 3.31 (2H, dd, *J* = 2.6, 1.5 Hz, H4), 6.39 (1H, s, H10′), 7.24 (s, CHCl_3_), 7.45 (2H, m, H16, H16′), 7.79 (1H, s, H10); ^13^C-NMR (CDCl_3_, 125 MHz) *δ* 18.1 (C-18 or C-18′), 18.3 (C-18′ or C-18), 21.0 (C-8 or C-8′), 21.3 (C-8′ or C-8), 25.7 (C-9 or C-9′), 25.9 (C9′ or C-9), 31.8 (C-4 or C-4′), 31.9 (C-4′ or C-4), 37.11 (C-7 or C-7′), 37.14 (C-7′ or C-7), 39.2 (C-6 or C-6′), 39.5 (C-6′ or C-6), 39.7 (C-1 or C-1′), 39.8 (C-1′ or C-1), 46.7 (C-5 or C-5′), 47.0 (C-5′ or C-5), 77.1 (CHCl_3_), 124.3 (C-11′), 125.5 (C-15 or C-15′), 125.6 (C-15′ or C-15), 126.2 (C-16 or C-16′), 126.3 (C-16′ or C-16), 127.2 (C-10′), 129.4 (C-11), 129.2 (C-10), 136.0 (C-17), 141.9 (C-2 or C-2′), 142.5 (C-2′ or C-2), 147.8 (C-14), 158.3 (C-13), 159.5 (C-12 or C-12′), 159.7 (C-12′ or C-12), 161.3 (C-3 or C-3′), 161.4 (C-3′ or C-3); EI MS *m*/*z* 945 [M]^+^ (3), 620 (2), 592 (2), 577 (4), 411 (20), 410 (54), 288 (35), 248 (100), 207 (35), 203 (80), 202 (59), 190 (58), 189 (63), 133 (93), 121 (41), 119 (62), 107 (52), 105 (45), 95 (57), 69 (52); HREIMS *m*/*z* 944.5502(calcd for C_66_H_68_N_6_^+•^, 944.5500).

*N1,N4-bis((1R,3R,8R,10R)-2,2,9,9-Tetramethyl-3,4,7,8,9,10-hexahydro-1H-1,3:8,10-dimethanocyclo-penta[1,2-b:5,4-b′]diquinolin-12(2H)-ylidene)benzene-1,4-diamine* (**4**). Solvate with CHCl_3_. Orange powder; ${\left[\alpha \right]}_{D}^{23}$ + 5.9 (*c* 0.37, CHCl_3_); ${\left[\alpha \right]}_{546}^{23}$ + 11 (*c* 0.37, CHCl_3_); UV (CHCl_3_) λ_max_ (lg ε) 426 (3.68), 339 (4.47), 333 (4.40), 324 (4.42), 280 (sh)(4.43), 252 (4.76) nm; IR (KBr) ν_max_ 3040-2850 (ν_C-H_), 1642 (ν_C=N_), 1597, 1571, 1555, 1490, 1470, 1424, 1395 (the most intensive band), 1261, 1215, 1100, 1071, 947, 927, 901, 847, 772, 752 (δ_Ar-H_), 732 cm^−1^; ^1^H NMR (CDCl_3_, 400 MHz) *δ* 0.59 (3H, s, H8 or H8′), 0.70 (3H, s, H8′ or H8), 1.16 (1H, d, *J* = 9.7 Hz, *pro*-*R*-H7′), 1.31 (1H, d, *J* = 9.7 Hz, *pro*-*R*-H7), 1.33 (3H, s, H9′), 1.42 (3H, s, H9), 2.30 (1H, dddd, *J* = 5.6, 5.6, 2.4, 2.4 Hz, H5′), 2.38 (1H, dddd, *J* = 5.6, 5.6, 2.4, 2.4 Hz, H5), 2.51 (1H, dd, *J* = 5.6, 5.6, H1′), 2.60 (1H, ddd, *J* = 9.7, 5.6, 5.6 Hz, *pro*-*S*-H7′), 2.71 (1H, ddd, *J* = 9.7, 5.6, 5.6 Hz, *pro*-*S*-H7), 2.85 (1H, dd, *J* = 5.6, 5.6 Hz, H1), 3.21 (2H, d, *J* = 2.4 Hz, H4′), 3.30 (2H, d, *J* = 2.6 Hz, H4), 6.67 (1H, s, H10′), 7.07 (2H, s, H15), 7.24 (s, CHCl_3_), 7.70 (1H, s, H10); ^13^C NMR (CDCl_3_, 125 MHz) *δ* 21.1 (C-8 or C-8′), 21.2 (C-8′ or C-8), 25.7 (C-9 or C-9′), 25.9 (C-9′ or C-9), 31.77 (C-4 or C-4′), 31.84 (C-4′ or C-4), 37.04 (C-7 or C-7′), 37.1 (C-7′ or C-7), 39.2 (C-6 or C-6′), 39.4 (C-6′ or C-6), 39.6 (C-1 or C-1′), 39.8 (C-1′ or C-1), 46.7 (C-5 or C-5′), 47.0 (C-5′ or C-5), 77.1 (CHCl_3_), 119.8 (C-15), 123.5 (C-11′), 127.1 (C-10′), 129.5 (C-11), 130.5 (C-10), 141.2 (C-2′), 142.5 (C-2), 147.7 (C-14), 158.2 (C-13), 159.9 (C-12 or C-12′), 160.0 (C-12′ or C-12), 161.4 (C-3 or C-3′), 161.6 (C-3′ or C-3); EI MS *m*/*z* 812 [M]^+^ (84), 798 (14), 797 (22), 769 (21), 460 (58), 445 (18), 356 (33), 355 (27), 354 (21), 341 (22), 208 (21), 207 (100). HREIMS *m*/*z* 812.4558 (calcd for C_56_H_56_N_6_^+•^, 812.4561).

*Crystal data*: C_56_H_56_N_6_+2(CHCl_3_), *M* = 1051.80, monoclinic, space group *C*2, at 110 K: *a* = 38.5594(17), *b* = 10.3080(3), *c* = 13.8452(6) Å, β = 97.298(2)°, *V* = 5458.5(4) Å^3^, *Z* = 4, *d*_calc_ = 1.280 g·cm^−3^, μ = 0.358 mm^−1^, a total of 26,868 (θ_max_ = 25.68°), 10,365 unique (*R*_int_ = 0.0560), 7852 [*I* > 2σ(*I*)], 639 parameters. GooF = 1.026, *R*_1_ = 0.0546, *wR*_2_ = 0.1122 [*I* > 2σ(*I*)], *R*_1_ = 0.0808, *wR*_2_ = 0.1248 (all data), max/min diff. peak 1.15/−0.46 e·Å^−3^. Two types of solvent accessible voids with volumes of 75 and 46 Å^3^ were found using PLATON [10,11]. Residual electron density peaks located in the voids (integral electron density of 3 and 4 e^−^) indicate additional small molecules (water, air, etc.) can be presented in the crystal structure.

*N1,N3-bis((1R,3R,8R,10R)-2,2,9,9-Tetramethyl-3,4,7,8,9,10-hexahydro-1H-1,3:8,10-dimethanocyclo-penta[1,2-b:5,4-b′]diquinolin-12(2H)-ylidene)benzene-1,3-diamine* (**5**). Solvate with CHCl_3_. Yellow powder; ${\left[\alpha \right]}_{D}^{23}$ − 190 (*c* 0.79, CHCl_3_); ${\left[\alpha \right]}_{546}^{23}$ − 260 (*c* 0.79, CHCl_3_); UV (CHCl_3_) λ_max_ (lg ε) 385 (ε 3.22), 339 (4.14), 332 (4.07), 324 (4.08), 282 (sh) (4.13) 258 (4.44), 250 (4.45) nm; IR (KBr) ν_max_ 3055-2850 (ν_C-H_), 1646 (ν_C=N_), 1597, 1584, 1573, 1555, 1471, 1424, 1395 (the most intensive band), 1326, 1259, 1214, 1183, 1143, 1099, 1072. 1052, 928, 905, 814, 772, 731 (δ_Ar-H_), 699 (δ_Ar-H_), 643, 543 cm^−1^; ^1^H-NMR (CDCl_3_,400 MHz) *δ* 0.61 (3H, s, H8 or H8′), 0.68 (3H, s, H8′ or H8), 1.18 (1H, d, *J* = 9.7 Hz, *pro*-*R*-H7′), 1.29 (1H, d, *J* = 9.7 Hz, *pro*-*R*-H7), 1.33 (3H, s, H9 or H9′), 1.40 (3H, s, H9′ or H9), 2.30 (1H, dddd, *J* = 5.6, 5.6, 2.4, 2.4 Hz, H5′), 2.37 (1H, dddd, *J* = 5.6, 5.6, 2.4, 2.4 Hz, H5), 2.47 (1H, dd, *J* = 5.6, 5.6 Hz, H1′), 2.61 (1H, ddd, *J* = 9.7, 5.6, 5.6 Hz, *pro*-*S*-H7′), 2.70 (1H, ddd, *J* = 9.7, 5.6, 5.6 Hz, *pro*-*S*-H7), 2.82 (1H, dd, *J* = 5.6, 5.6 Hz, H1), 3.21 (2H, d, *J* = 2.4 Hz, H4′), 3.28 (2H, d, *J* = 2.6 Hz, H4), 6.56 (1H, s, H10′), 6.68 (0.5H, t, *J* = 1.9 Hz, H15), 6.82 (1H, dd, *J* = 7.8, 1.9 Hz, H16), 7.24 (s, CHCl_3_), 7.41 (0.5H, t, *J* = 7.8 Hz, H17), 7.66 (1H, s, H10); ^13^C-NMR (CDCl_3_, 125 MHz) *δ* 21.1 (C-8 or C-8′), 21.2 (C-8′ or C-8), 25.7 (C-9 or C-9′), 25.9 (C-9′ or C-9), 31.8 (C-4and C-4′), 37.0 (C-7 or C-7′), 37.1 (C-7′ or C-7), 39.3 (C-6 or C-6′), 39.4 (C-6′ or C-6), 39.6 (C-1 or C-1′), 39.8 (C-1′ or C-1), 46.8 (C-5 or C-5′), 46.9 (C-5′ or C-5), 77.1 (CHCl_3_), 108.8 (C-15), 114.4 (C-16), 123.3 (C-11′), 127.2 (C-10′), 129.4 (C-11), 130.0 (C-17), 130.9 (C-10), 141.2 (C-2 or C-2′), 142.5 (C-2′ or C-2), 152.3 (C-14), 158.3 (C-13), 159.96 (C-12 or C-12′), 160.03 (C-12′ or C-12), 161.4 (C-3 or C-3′), 161.7 (C-3′ or C-3); EI MS *m*/*z* 812 [M]^+^ (100), 797 (19), 769 (27), 406 (11), 370 (28), 355 (20), 207 (25), 121 (27), 84 (21), 77 (15); HREIMS *m*/*z* 812.4568 (calcd for C_56_H_56_N_6_^+•^, 812.4561).

*N1,N2-bis((1R,3R,8R,10R)-2,2,9,9-Tetramethyl-3,4,7,8,9,10-hexahydro-1H-1,3:8,10-dimethanocyclo-penta[1,2-b:5,4-b′]diquinolin-12(2H)-ylidene)benzene-1,2-diamine* (**6**). Solvate with CHCl_3_. Orange powder; ${\left[\alpha \right]}_{D}^{23}$ − 224 (*c* 0.691, CHCl_3_); ${\left[\alpha \right]}_{546}^{23}$ − 387 (*c* 0.691, CHCl_3_); UV (CHCl_3_) λ_max_ (lg ε) 400 (3.39), 339 (4.54), 333 (4.48), 324 (4.49), 257 (4.81), 251 (4.81) nm; IR (KBr) ν_max_ 3054-2850 (ν_C-H_), 1646 (ν_C=N_), 1597, 1572, 1554, 1470, 1440, 1424, 1395 (the most intensive band), 1262, 1214, 1182, 1153, 1102, 1071, 1052, 1034, 947, 926, 822, 772, 753 (δ_Ar-H_), 727, 705 cm^−1^; ^1^H-NMR (CDCl_3_,300 MHz) *δ* 0.51 (3H, s, H8 or H8′), 0.57 (3H, s, H8′ or H8), 1.14 (1H, d, *J* = 9.7 Hz, *pro*-*R*-H7 or *pro*-*R*-H7′), 1.15 (1H, d, *J* = 9.7 Hz, *pro*-*R*-H7′ or *pro*-*R*-H7), 1.27 (3H, s, H9 or H9′), 1.28 (3H, s, H9′ or H9), 2.24 (2H, dddd, *J* = 4.2, 4.2, 2.4, 2.4 Hz, H5 and H5′), 2.44 (1H, dd, *J* = 4.2, 4.2 Hz, H1′), 2.55 (2H, ddd, *J* = 9.7, 4.2, 4.2 Hz, *pro*-*S*-H7 and *pro*-*S*-H7′), 2.60 (1H, dd, *J* = 4.2, 4.2 Hz, H1), 3.15 (4H, m, H4 and H4′), 6.79 (1H, s, H10′), 6.97 (1H, dd, *J* = 4.4, 2.6 Hz, H15), 7.19 (1H, dd, *J* = 4.4, 2.6 Hz, H16), 7.24 (s, CHCl_3_), 7.28 (1H, s, H10); ^13^C-NMR (CDCl_3_, 75 MHz) *δ* 21.0 (C-8 or C-8′), 21.1 (C-8′ or C-8), 25.7 (C-9 or C-9′), 25.8 (C-9′ or C-9), 31.7 (C-4 or C-4′), 31.9 (C-4′ or C-4), 36.98 (C-7 or C-7′), 37.01 (C-7′ or C-7), 39.21 (C-6 or C-6′), 39.23 (C-6′ or C-6), 39.63 (C-1 or C-1′), 39.66 (C-1′ or C-1), 46.7 (C-5 or C-5′), 46.8 (C-5′ or C-5), 77.1 (CHCl_3_), 119.9 (16), 123.9 (C-11′), 125.2 (C-15), 127.1 (C-101), 129.5 (C-11), 130.7 (C-10), 140.1 (C-14), 141.2 (C-2 or C-2′), 142.2 (C-2′ or C-2), 158.0 (C-13), 159.4 (C-12 or C-12′), 159.8 (C-12′ or C-12), 161.0 (C-3 or C-3′), 161.2 (C-3′ or C-3); EI MS *m*/*z* 812 [M]^+^ (100), 797 (10), 769 (9), 520 (37), 489 (66), 481 (10), 380 (12), 355 (11), 230 (14), 207 (47), 180 (16), 172 (48), 168 (27), 118 (18), 109 (16), 91 (46), 84 (14), 82 (30); HREIMS *m*/*z* 812.4568 (calcd for C_56_H_56_N_6_^+•^, 812.4561).

*Crystal data*: C_56_H_56_N_6_ + 4(CHCl_3_), M = 1290.54, monoclinic, space group *C*2, at 296 K: *a* = 21.5995(10), *b* = 10.5615(6), *c* = 15.4531(8) Å, β = 113.727(2)°, *V* = 3227.2(3) Å^3^, *Z* = 2, *d*_calc_ = 1.328 g·cm^−3^, μ = 0.557 mm^−1^, a total of 33,594 (θ_max_ = 25.36°), 5877 unique (*R*_int_ = 0.0447), 5088 [*I* > 2σ(*I*)], 354 parameters. GooF = 1.028, *R*_1_ = 0.0584, *wR*_2_ = 0.1509 [*I* > 2σ(*I*)], *R*_1_ = 0.0679, *wR*_2_ = 0.1605 (all data), max/min diff. peak 0.463/−0.409 e·Å^−3^.

*N1,N5-bis((1R,3R,8R,10R)-2,2,9,9-Tetramethyl-3,4,7,8,9,10-hexahydro-1H-1,3:8,10-dimethanocyclo-penta[1,2-b:5,4-b′]diquinolin-12(2H)-ylidene)naphthalene-1,5-diamine* (**7**). Solvate with 1,4-dioxane. Orange powder; ${\left[\alpha \right]}_{D}^{23}$ − 249 (*c* 0.053, CHCl_3_); ${\left[\alpha \right]}_{546}^{23}$ − 427 (*c* 0.053, CHCl_3_); UV (CHCl_3_) λ_max_ (lg ε) 442 (3.69), 339 (4.61), 332 (4.60), 324 (4.64), 250 (4.90) nm; IR (KBr) ν_max_ 3040-2850 (ν_C-H_), 1645 (ν_C=N_), 1597, 1572, 1555, 1501, 1470, 1424, 1395 (the most intensive band), 1258, 1214, 1183, 1100, 1071, 1053, 947, 926, 915, 810, 784, 773, 731 cm^−1^; ^1^H-NMR (CDCl_3_,400 MHz) *δ* 0.53 (3H, s, H8 or H8′), 0.77 (3H, s, H8′ or H8), 1.09 (1H, d, *J* = 9.7 Hz, *pro*-*R*-H7′), 1.27 (3H, s, H9 or H9′), 1.36 (1H, d, *J* = 9.7 Hz, *pro*-*R*-H7), 1.46 (3H, s, H9′ or H9), 2.27 (2H, m, H5′ and H1′), 2.42 (1H, dddd, *J* = 5.7, 5.7, 2.4, 2.4 Hz, H5), 2.52 (1H, ddd, *J* = 9.7, 5.7, 5.7 Hz, *pro*-*S*-H7′), 2.76 (1H, ddd, *J* = 9.7, 5.7, 5.7 Hz, *pro*-*S*-H7), 2.91 (1H, dd, *J* = 5.7, 5.7 Hz, H1), 3.18 (2H, d, *J* = 2.4 Hz, H4′), 3.34 (2H, d, *J* = 2.6 Hz, H4), 3.68 (s, 1,4-dioxane), 6.30 (1H, s, H10′), 7.03 (1H, dd, *J* = 7.2, 0.7 Hz, H15), 7.39 (1H, dd, *J* = 8.4, 7.2 Hz, H16), 7.75 (1H, dd, *J* = 8.4, 0.7 Hz, H17), 7.84 (1H, s, H10); ^13^C-NMR (CDCl_3_, 75 MHz) *δ* 21.1 (C-8 or C-8′), 21.3 (C-8′ or C-8), 25.7 (C-9 or C-9′), 26.0 (C-9′ or C-9), 31.8 (C-4 or C-4′), 32.0 (C-4′ or C-4), 37.1 (C-7 or C-7′), 37.3 (C-7′ or C-7), 39.2 (C-6 or C-6′), 39.5 (C-6′ or C-6), 39.8 (C-1 or C-1′), 40.0 (C-1′ or C-1), 46.7 (C-5 or C-5′), 47.2 (C-5′ or C-5), 67.0 (1,4-dioxane), 113.9 (C-15), 120.3 (C-16), 123.6 (C-11′), 125.7 (C-17), 126.1 (C-18), 127.3 (C-10′), 129.6 (C-11), 130.3 (C-10), 141.3 (C-2 or C-2′), 142.5 (C-2′ or C-2), 147.4 (C-14), 158.5 (C-13), 160.2 (C-12 or C-12′), 160.4 (C-12′ or C-12), 161.4 (C-3 or C-3′), 161.8 (C-3′ or C-3); EI MS *m*/*z* 862 [M]^+^ (17), 510 (15), 369 (27), 370 (18), 355 (8), 281 (16), 208 (9), 256 (10), 244 (21), 209 (12), 208 (18), 207 (81), 91 (20), 84 (16). HREIMS *m*/*z* 862.4722 (calcd for C_60_H_58_N_6_^+•^, 862.4717).

*1,2-bis((1R,3R,8R,10R)-2,2,9,9-Tetramethyl-3,4,7,8,9,10-hexahydro-1H-1,3:8,10-dimethanocyclo-penta[1,2-b:5,4-b′]diquinolin-12(2H)-ylidene)hydrazine* (**8**). Solvate with CHCl_3_ and C_6_H_6_. Red powder; ${\left[\alpha \right]}_{D}^{25}$ − 146 (*c* 1.04, CHCl_3_); ${\left[\alpha \right]}_{546}^{25}$ − 220 (*c* 1.04, CHCl_3_); UV (CHCl_3_) λ_max_ (lg ε) 442 (sh) (3.34), 395 (4.16), 376 (4.40), 359 (4.41), 342 (4.71), 335 (4.65), 326 (4.68) nm; IR (KBr) ν_max_ 3036-2850 (ν_C-H_), 1625 (ν_C=N_), 1596, 1560, 1554, 1470, 1424, 1392 (the most intensive band), 1369, 1281, 1268, 1214, 1181, 1100, 1071, 1034, 1017, 946, 927, 807, 769, 752, 678 (C_6_H_6_) cm^−1^; ^1^H-NMR (CDCl_3_, 500 MHz) *δ* 0.64 (3H, s, H8), 0.71 (3H, s, H8′), 1.27 (1H, d, *J* = 9.7 Hz, *pro*-*R*-H7), 1.34 (1H, d, *J* = 9.7 Hz, *pro*-*R*-H7′), 1.39 (3H, s, H9), 1.43 (3H, s, H9′), 2.34 (1H, ddddd, *J* = 5.6, 5.6, 2.4, 2.4, 1.2 Hz, H5), 2.39 (1H, ddddd, *J* = 5.6, 5.6, 2.4, 2.4, 1.2 Hz, H5′), 2.68 (1H, dddd, *J* = 9.7, 5.6, 5.6, 1.5 Hz, *pro*-*S*-H7), 2.74 (1H, dddd, *J* = 9.7, 5.6, 5.6, 1.5 Hz, *pro*-*S*-H7′), 2.76 (1H, dd, *J* = 5.6, 5.6 Hz, H1), 2.92 (1H, dd, *J* = 5.6, 5.6 Hz, H1′), 3.25 (2H, dd, *J* = 2.4, 1.5 Hz, H4), 3.29 (2H, dd, *J* = 2.6, 1.5 Hz, H4′), 7.24 (s, CHCl_3_), 7.32 (s, benzene), 7.71 (1H, s, H10), 8.02 (1H, s, H10′); ^13^C-NMR (CDCl_3_, 125 MHz) *δ* 21.1 (C-8 or C-8′), 21.2 (C-8′ or C-8), 25.82 (C-9 or C-9′), 25.88 (C-9′ or C-9), 31.85 (C-4 or C-4′), 31.88 (C-4′ or C-4), 37.05 (C-7 or C-7′), 37.08 (C-7′ or C-7), 39.38 (C-6 or C-6′), 39.44 (C-6′ or C-6), 39.7 (C-1 or C-1′), 39.8 (C-1′ or C-1), 47.0 (C-5 or C-5′), 47.1 (C-5′ or C-5), 77.1 (CHCl_3_), 123.7 (C-11′), 126.9 (C-10′), 128.2 (benzene), 128.6 (C-11), 134.0 (C-10), 141.7 (C-2 or C-2′), 142.2 (C-2′ or C-2), 155.4 (C-13), 157.8 (C-12 or C-12′), 158.7 (C-12′ or C-12), 161.0 (C-3 or C-3′), 161.1 (C-3′ or C-3); EI MS *m*/*z* 736 [M]^+^ (100), 721 (17), 369 (27), 370 (29), 355 (19), 327 (14). 281 (11), 208 (9), 207 (43), 83 (10). HREIMS *m*/*z* 736.4254 (calcd for C_50_H_52_N_6_^+•^, 736.4248).

*Crystal data*: C_50_H_52_N_6_+solvent, *M* = 736.98, monoclinic, space group *C*2, at 296 K: *a* = 22.481(8), *b* = 7.638(2), *c* = 20.367(7) Å, β = 120.44(2)°, *V* = 3014.7(19) Å^3^, *Z* = 2, *d*_calc_ = 0.812 g·cm^−3^, μ = 0.048 mm^−1^, a total of 23,835 (θ_max_ = 25.1°), 4850 unique (*R*_int_ = 0.125), 2104 [*I* > 2σ(*I*)], 257 parameters. GooF = 0.90, *R*_1_= 0.0726, *wR*_2_ = 0.1749 [*I* > 2s(*I*)], *R*_1_ = 0.1529, *wR*_2_ = 0.1749 (all data), max/min diff. peak 0.216/−0.221 e·Å^−3^.

*(1′R,3′R,8′R,10′R)-2′,2′,9′,9′-Tetramethyl-1′,2′,3′,4′,7′,8′,9′,10′-octahydro-1H,3H-spiro[perimidine-2,12′-[1,3:8,10]dimethanocyclopenta[2,1-b:3,4-b′]diquinoline]* (**9**). Perchloric acid (0.1 mmol, 15 µL of 72% aqueous solution) was added to a solution of pinodiazafluorenone 1 (1 mmol, 370 mg) and 1,8-diaminohaphtalene (1 mmol, 158 mg) in ethanol (3 mL). The resulting solution was refluxed under Ar for 5 h. The reaction mixture was cooled down to r.t. and the solvent was removed under reduced pressure. The residue was taken up in benzene (5 mL), the solution was washed with aqueous oxalic acid (1 mL, 10% in H_2_O) and evaporated in vacuum. The crude product was percolated through a silica gel column (eluent benzene-chloroform 10:1). The eluate was concentrated under reduced pressure, and the residual solvent was removed by heating to 100 °C at reduced pressure (2–3 mm Hg) to produce the title product (357 mg, yield 80%) as a glassy substance, which is rapidly oxidized in open air. ${\left[\alpha \right]}_{D}^{20}$ − 74.0 (c 0.289, CHCl_3_); UV (CHCl_3_) λ_max_ (lg ε) 346 (sh) (3.91), 329 (4.05), 286 (3.70), 243 (4.04) nm; IR (KBr) ν_max_ 3390 (asym. ν_N-H_), 3349 (sym. ν_N-H_), 3055-2850 (ν_C-H_), 1624, 1599, 1561, 1468, 1423, 1398 (the most intensive band), 1322, 1279, 1252, 1214, 1182, 1118, 1069, 947, 920, 813 (δ_Ar-H_), 759 (δ_Ar-H_)cm^−1^; ^1^H-NMR (CDCl_3_,400 MHz) *δ*0.63 (3H, s, H8), 1.19 (1H, dd, *J* = 8.61, 4.2 Hz, *pro*-*R*-H7), 1.33 (3H, s, H9), 2.33 (1H, ddddd, *J* = 5.6, 5.6, 2.4, 2.4, 1.2, Hz, H5′), 2.39 (1H, ddddd, *J* = 5.6, 5.6, 2.4, 2.4, 1.2 Hz, H5), 2.60 (2H, m, *pro*-*S*-H7, H1), 3.24 (2H, m, *W*_1/2_ = 7 Hz, H4), 4.55 (1H, s, NH), 6.52 (1H, dd, *J* = 2.1, 6.2 Hz, H15), 7.03 (1H, s, H10), 7.27 (2H, m, H16, H17); ^13^C-NMR (CDCl_3_, 100 MHz) *δ*21.2 (C-8), 25.8 (C-9), 31.9 (C-4), 36.8 (C-7), 39.3 (C-6), 39.9 (C-1), 46.9 (C-5), 71.4 (C-13), 107.0 (C-15), 112.5 (C-19), 118.1 (C-17), 127.1 (C-16), 128.2 (C-10), 134.3 (C-18), 139.5 (C-11 or C-14), 139.8 (C-11 or C-14), 142.3 (C-2), 154.9 (C-12), 159.6 (C-3); EI MS *m*/*z* 510 [M]^+^ (100%), 495 (6), 469 (4), 467 (5), 453 (4), 400 (16), 370 (15), 355 (7), 327 (5), 329 (5), 285 (9), 279 (6), 278 (7), 173 (6), 134 (5), 91 (13), 78 (31), 45 (21); HREIMS *m*/*z* 510.2781 (calcd for C_35_H_34_N_4_^+•^, 510.2778).

*6-Chloro-N1-((1R,3R,8R,10R)-2,2,9,9-tetramethyl-3,4,7,8,9,10-hexahydro-1H-1,3:8,10-dimethano-cyclopenta[1,2-b:5,4-b′]diquinolin-12(2H)-ylidene)naphthalene-1,5-diamine* (**11**). ^1^H-NMR (CDCl_3_,400 MHz) *δ* 0.53 (3H, s, H8 or H8′), 0.74 (3H, s, H8′ or H8), 1.08 (1H, d, *J* = 9.7 Hz, *pro*-*R*-H7′), 1.25 (s, 3H, H9 or H9′), 1.33 (1H, d, *J* = 9.7 Hz, *pro*-*R*-H7), 1.43 (3H, s, H9′ or H9), 1.99 (3H, s, CH_3_CN), 2.25 (2H, m, H5′ and H1′), 2.40 (1H, dddd, *J* = 5.7, 5.7, 2.4, 2.4 Hz, H5), 2.51 (1H, ddd, *J* = 9.7, 5.7, 5.7 Hz, *pro*-*S*-H7′), 2.73 (1H, ddd, *J* = 9.7, 5.7, 5.7 Hz, *pro*-*S*-H7), 2.89 (1H, dd, *J* = 5.7, 5.7 Hz, H1), 3.16 (2H, d, *J* = 2.4 Hz, H4′), 3.31 (2H, d, *J* = 2.6 Hz, H4), 4.63 (2H, br.s *W*_1/2_ = 5 Hz, NH_2_), 6.16 (1H, s, H10′), 6.96 (1H, dd, *J* = 7.7, 1.0 Hz, H14), 7.23 (1H, d, *J* = 9.0 Hz, *W*_1/2_ = 3 Hz, H17), 7.26 (1H, d, *J* = 9.0 Hz, H18), 7.47 (1H, dd, *J* = 8.5, 7.7 Hz, H15), 7.65 (1H, ddd, *J* = 8.5, 1.0, 0.7 Hz, H16), 7.80 (1H, s, H10); EI MS *m*/*z* 544 [M]^+^ (100), 529 (37), 510 (18), 503 (21), 459 (10), 370 (7), 355 (5), 284 (6), 256 (11), 207 (19), 192 (11), 129 (10), 97 (15), 83 (14), 69 (25), 57 (31), 55 (35), 44 (59); HREIMS *m*/*z* 544.2390 (calcd for C_35_H_33_ClN_4_^+•^, 544.2388).

*Crystal data*: C_35_H_33_ClN_4_+C_2_H_3_N, *M* = 586.16, monoclinic, space group *P*2_1_, at 150 K: *a* = 9.9004(8), *b* = 14.5959(16), *c* = 11.1771(10) Å, β = 110.729(2)°, *V* = 1510.6(2) Å^3^, *Z* = 2, *d*_calc_ = 1.289 g·cm^−3^, μ = 0.162 mm^−1^, a total of 11,506 (θ_max_ = 25.90°), 5437 unique (*R*_int_ = 0.0503), 4015 [*I* > 2σ(*I*)], 401 parameters. GooF = 1.022, *R*_1_ = 0.0603, *wR*_2_ = 0.1343 [*I* > 2σ(*I*)], *R*_1_ = 0.0919, *wR*_2_ = 0.1512 (all data), max/min diff. peak 0.47/-0.48 e·Å^−3^.

### 3.5. Supplementary X-Ray Crystallographic Data

CCDC 1,937,267 (**4**), 1,935,347 (**6**), 1,935,346 (**8**) and 1,937,837 (**11**) contain the supplementary crystallographic data for this paper. These data can be obtained free of charge from The Cambridge Crystallographic Data Centre via http://www.ccdc.cam.ac.uk/conts/retrieving.html (or from the CCDC, 12 Union Road, CambridgeCB2 1EZ, UK; Fax: +44-1223-336033; E-mail: deposit@ccdc.cam.ac.uk).

### 3.6. Supplementary NMR Data

The Appendix A contains ^1^H- and ^13^C-NMR spectra of the compounds **2**–**9**, **11**.

## 4. Conclusions

The chiral dipinofluorenone ((1*R*,3*R*,8*R*,10*R*)-2,2,9,9-tetramethyl-3,4,7,8,9,10-hexaydro-1*H*-1,3:8,10-dimethanocyclopenta[1,2-*b*:5,4-*b*′]diquinolin-12(2*H*)-one) (**1**) is a convenient starting compound for the synthesis of a new group of chiral bisimines by condensation with primary diamines. The traditional method of the imines preparation (boiling a mixture of amine and ketone in acetic acid) does not provide satisfactory results, giving low yields of the desired products. Higher yields can be obtained by boiling a mixture of dipinofluorenone and a diamine in aromatic hydrocarbons (benzene, toluene, xylene) in the presence of TsOH with azeotropic distillation of the water formed, and the yields increase with increasing solvent boiling point. The best yields are obtained when condensation of dipinofluorenone and a diamine is carried out in the system TiCl_4_–Py. In contrast to bis-imines derived from 4,5-diazafluoren-9-one, bisimines of the pinodiazafluorene series **2**–**8** are fairly stable compounds. Bisimines **2**–**8**, due to the peculiarities of the structure of their molecules, cannot provide the closest packing by crystallization of the molecules by themselves, and as a result, bisimines **2**–**8** form crystals with a large volume of free cavities, which are filled with solvent molecules (organic solvents and water), the solvates being quickly lost the solvent when exposed to air. Bisimines of the pinodiazafluorene series **2**–**8** give color reactions with transition metal ions and can be regarded as prospective polydentate ligands with interesting luminescent and chiroptical properties.

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
