# Peer review of "Chiral C2-Symmetric Diimines with 4,5-Diazafluorene Units"

_molecules, 2019, doi:10.3390/molecules24173186_

Round 1

Reviewer 1 Report

Peer-Review of Molecules Manuscript 564536

The manuscript entitled “Chiral C2- symmetric diimines with 4,5-diazafluorene unit” describes the preparation and characterization of some new interesting chiral diimines.

The work leading to these compounds is mostly well presented, accurately supported by the included references, and conclusions are in line with results obtained.

I only have a few minor comments that I think can improve the submitted manuscript before publication in Molecules is granted.

1.- The compounds listed in Figure 1 can easily be resumed by considering the repeated dipinodiazafluorene moieties as substituents of the different central diamines used. This should simplify the figure.

2.- In the abstract, the authors describe the prepared compounds as having interesting chiroptical properties. This is not explained in the manuscript any further that pointing out the differences in optical rotations. Perhaps these differences are in part related to the appearance of inherently dissymmetric chromophores in some of the prepared compounds.

3.- In the manuscript is described the lost in magnetic equivalency between the two pinopyridine moieties from the ketone to the Schiff base. I also think is worth mentioning that unlike other diimines from complex ketones, the new double bond did not give a pair of distereoisomers.

Author Response

The molecules were redrawn according to the reviewer recommendations. At the moment we have no facilities to register CD spectra. Instead of CD spectra, we added optical rotation at 546 nm (and the corresponding coment). Significant differences in optical rotations at 589 nm (D line) and 546 nm for all the compounds studied indicate the presence of optically active bands in the visible part of the spectrum (violet and blue).

Starting pinodiazofluorenone 1 belongs to the symmetry point group C2 with rotational axis C2 coinciding with the bond C=O. For this reason, the formation of imines does not produce pairs of distereoisomers (as well as in the case of acetone, bezophenone, 4,5-diazafluoren-9-one, etc.)

Reviewer 2 Report

The paper describes a synthetic procedure for the preparation of a new group of chiral C2‐symmetric diimines with the 4,5‐diazafluorene core, starting from dipinodiazafluorene and corresponding diamines. The products were analyzed and confirmed using usual methods, and some of them were analyzed with X-ray. Three compounds exhibited photoluminescence.

The compounds described seem interesting and potentially have useful applications. However, a couple of things should be addressed before the publication of these results.

Major issues:

1.       In the abstract and the conclusion it is stated that “Bis‐imines of the pinodiazafluorene series 2–8 give color reactions with transition metal ions”. This seems really interesting, but no evidence of this behavior is presented in the paper. If such experiments were conducted, it should be added to the paper or this statement should be deleted. It would be a nice supplement to the photoluminescence part which is written too vague and too short in this version of the paper.

2.       “Preparation of the compounds” part of the paper is not well presented. A couple of methods were tried, but it is hard to see any straightforward way through the optimization process and the choice of methods. I propose that Scheme 1 and Figure 1 are combined in 1 scheme/figure and additional table is added with all the reaction conditions and yields (including the reactions that did not work and the yield is possibly 0).

3.       Since these compounds show interesting values of optical activity measured at 589 nm and possess chromophores in the structure, it would be of great interest to record CD spectra of these compounds.

Minor issues:

1.       It would be interesting to add a scheme for the preparation of compound 9. As well as the explanation for which reaction procedures were tried (just 1 sentence) and why the HClO4-EtOH procedure was chosen in the end.  

2.       Structure 10 can be incorporated in the same scheme since it seems its synthesis was tried under the same conditions. Which product was obtained in this reaction with ortho‐phenlinediamine?

3.       Please add NMR spectra of compounds (in supporting materials) since you base the purity of compounds on NMR.

4.       Compound 11 seems interesting, but the paragraph regarding it (page 4 104-113) should be moved after the X-ray data of prepared compounds 2-8. This way it is inserted in the middle of NMR/IR/X-ray analysis of prepared compounds. Also, were other methods for determining the yield of this compound tried (for example HPLC)?

5.       In synthetic procedures section, please add the ratio of benzene-chloroform eluent used for chromatography.

Author Response

"Major issues":

1. New table was added to the experimental section with the description of color effects.

2. Scheme 1 and Figure 1 were combined, the new table with the yields was added.

3. At the moment we have no facilities to register CD spectra. Instead of CD spectra, we added optical rotation at 546 nm (and the corresponding coment). Significant differences in optical rotations at 589 nm (D line) and 546 nm for all the compounds studied indicate the presence of optically active bands in the visible part of the spectrum (violet and blue).

"Minor issues":

1. New scheme was added as well as additional explanations of formation of compound 9.

2. Structure 10 was incorporated to the new scheme.

3. NMR spectra were added as Supplementary Data.

4. X-Ray data of all the compounds (desired products 4,6,8 and artefact 11) have many things in common, therefore all the X-ray data are discussed together. In this regard, it seems illogical if compound 11 appeared after discussion on x-ray data. Thus, discussion of compound 11 was left unchanged. 

4a. We did not try other methods (like HPLC) for determining the yield of compound 11. The upper yield limit is estimated at 0.5% by 1H NMR, and more precise value does not make any sense yet.

5. Ratio of benzene-chloroform eluent was added.

Reviewer 3 Report

The manuscript needs to be proofreading by a native English speaking before to be reviewed.

Therefore, I'm not able to provide any recommendation.

Author Response

The manuscript was corrected by a professional translator.

Round 2

Reviewer 2 Report

The authors responded to all the comments and reviewed the manuscript according to them. Just a few editing changes are required.

In Scheme 1 below reaction please add the label "X:" or "X=" to denote structure moieties more clearly. Also, yields can be removed since they are now in the table. Number all the figures in the paper (two of them are not numbered in this version). Move Table 2 to the main text and add a sentence to explain its importance (to briefly explain why these compounds can be "regarded as prospective polydentate ligands with interesting luminescent and chiroptical properties" which you state in the conclusion). Given that this is primarily a synthetic paper, CD measurements are not crucial. However, if you plan to continue with the investigation of chirooptical properties, CD measurements will become a necessity. 

Author Response

(1) Scheme 1 has been modified (the label =N-X-N= is added, the yields are removed).
(2) All the figure are numbered (and renumbered when necessary).
(3) Tabl2 2 has been moved to the main text, the corresponding explanations are added.

Reviewer 3 Report

I recommended to revise the writing; however, the authors did not follow my recommendation. I attached the revised manuscript with some corrections (colored in yellow). Please, do the corrections and ask for help.

Author Response

Thank you for the comments. We have made all the corrections of the text fragments colored in yellow.